# Self-Exclusion among Online Poker Gamblers: Effects on Expenditure in Time and Money as Compared to Matched Controls

**DOI:** 10.3390/ijerph16224399

**Published:** 2019-11-11

**Authors:** Amandine Luquiens, Aline Dugravot, Henri Panjo, Amine Benyamina, Stéphane Gaïffas, Emmanuel Bacry

**Affiliations:** 1Hôpital Paul Brousse, APHP Villejuif, France. Université Paris-Saclay, 94800 Villejuif, France; amine.benyamina@aphp.fr; 2CMAP, Ecole Polytechnique, 91128 Palaiseau Cedex, France; stephane.gaiffas@polytechnique.edu (S.G.); emmanuel.bacry@polytechnique.edu (E.B.); 3Université Paris-Sud membre de Université Paris-Saclay, UVSQ, CESP, INSERM, 94800 Villejuif, France; aline.dugravot@inserm.fr (A.D.); henri.panjo@inserm.fr (H.P.); 4Addictions Department, Nîmes University Hospital, 30000 Nîmes, France; 5LPSM UMR 8001, Université Paris Diderot, 8001 Paris, France; 6CEREMADE UMR 7534 CNRS, Université Paris-Dauphine PSL, 75765 Paris Cedex 16, France

**Keywords:** online gambling, self-exclusion, responsible gambling, comparative study, poker

## Abstract

*Background:* No comparative data is available to report on the effect of online self-exclusion. The aim of this study was to assess the effect of self-exclusion in online poker gambling as compared to matched controls, after the end of the self-exclusion period. *Methods:* We included all gamblers who were first-time self-excluders over a 7-year period (*n* = 4887) on a poker website, and gamblers matched for gender, age and account duration (*n* = 4451). We report the effects over time of self-exclusion after it ended, on money (net losses) and time spent (session duration) using an analysis of variance procedure between mixed models with and without the interaction of time and self-exclusion. Analyzes were performed on the whole sample, on the sub-groups that were the most heavily involved in terms of time or money (higher quartiles) and among short-duration self-excluders (<3 months). *Results:* Significant effects of self-exclusion and short-duration self-exclusion were found for money and time spent over 12 months. Among the gamblers that were the most heavily involved financially, no significant effect on the amount spent was found. Among the gamblers who were the most heavily involved in terms of time, a significant effect was found on time spent. Short-duration self-exclusions showed no significant effect on the most heavily involved gamblers. *Conclusions:* Self-exclusion seems efficient in the long term. However, the effect on money spent of self-exclusions and of short-duration self-exclusions should be further explored among the most heavily involved gamblers.

## 1. Introduction

Harmful gambling behaviors are widespread and treatment-seeking is still very low among problem gamblers. Self-exclusion processes could be seen as an accessible tool for problem gamblers who are not ready to seek treatment. In France, gamblers can apply for online self-exclusion per website, for the length of their choice from one week to three years. During this period, they cannot access their gambling account on the website and receive no commercial offer from the gambling service provider. At the end of the period, they can gamble back on the website with no additional procedure. No help is provided nor any counselling during the self-exclusion period.

It has been consistently demonstrated that most self-excluders were indeed heavy gamblers and probably problem gamblers [1,2]. A recent meta-analysis describing gamblers who self-excluded highlighted that this tool, perceived as one of the main "responsible gambling" tools, is still considerably under-used [3]. Main barriers for self-exclusion has been described: complicated enrollment processes, lack of complete exclusion from all venues, little support from venue staff, and lack of adequate information on self-exclusion programs. The proportion of self-excluders could be particularly low among problem internet gamblers [4]. Regulators have reached the conclusion that this tool should be promoted to increase its use. Promoting the use of a potentially therapeutic tool needs to rely on robust efficacy data and not only on empirical data or mere common sense. A recent systematic review of the literature demonstrated that the impact of responsible gambling tools is still poorly supported by scientific evidence [5]. In particular, efficacy data for the effect of self-exclusion on gambling behaviors remains scarce [5]: several studies have shown reduced gambling after a self-exclusion period on both online and offline environments, with variable durations of follow-up, sometimes including the self-exclusion period itself [1,6,7]. Follow-up after online self-exclusion has been reported in only two studies [7,8]. The first one included a limited sample of 20 gamblers, with no control group, and assessed psychosocial outcomes [8]. The other one reported that the majority of online self-excluders returned to gambling after the self-exclusion period expired (*n* = 1996) [7], and that most of them self-excluded a second time, after another period of more intensive gambling than the first [1]. Only one study has reported follow-up data from matched controls with a comparative research design (*n* = 86) [5]. One study reported that gambling outcomes did not differ between self-exclusion alone vs self-exclusion combined with counseling or counseling only [9]. No study has reported efficacy data on spontaneous voluntary self-exclusion as compared to no intervention (i.e., no self-exclusion). One experimental study randomized volunteering problem gamblers (but not pathological gamblers, who were excluded) to either a very short 7-day period of self-exclusion or no self-exclusion [10]. The authors reported no significant between-group differences in terms of changes regarding money and time spent gambling at two months.

Skills are important in poker gambling and poker gamblers have been demonstrated to have particular thoughts about their own gambling behavior and to be particularly sensitive to feedbacks on their own practice [11]. Illusion of control could be high in poker gamblers [12] and perception of their own skills could be amplified [13]. The prevalence of problem gambling among online poker gamblers is particularly high, consistently reported between 15% and 20% of active gamblers [14,15]. Several factors predicting excessive gambling in poker gamblers were identified: stress, internal attribution, dissociation, boredom, negative emotions, irrational beliefs, anxiety, and impulsivity [16], lower performance in the emotional intelligence competences (Emotional Quotient inventory Short) and, in particular, those grouped in the Intrapersonal scale (emotional self-awareness, assertiveness, self-regard, independence and self-actualization). Classical financial moderators‘ relevance have been consistently discussed in poker gambling as financial involvement of problem gamblers can be very low and time involved is critical to take into account [14]. No time moderator is mandatory for online gamblers in France [17]. Self-exclusion could then be one of the most relevant tools currently available for poker online gamblers in France. Poker gamblers are then a particularly interesting population to study to assess the efficacy of self-exclusion.

The aim of the present study was to document the long-term effects of self-exclusion from a poker website as compared to no self-exclusion, using matched controls. Our hypothesis was that self-exclusion would have an effect on time and money spent after the exclusion ended compared to no self-exclusion. 

## 2. Methods

### 2.1. Population

We included all gamblers who self-excluded for the first time over a 7-year period from June 2010 up to October 2016 (*n* = 4887) on a poker website, Winamax ®, and1:1 matched gamblers who had never self-excluded up to the time of data collection, matched for gender, age and account duration automatically extracted from the account database following a structured query language (SQL) request. For technical reasons we could not match gamblers for the level of gambling involvement in terms of money/time, which were constructed variables not available from the SQL database. From the matched control group, we removed doubloons where one and the same gambler was matched to several self-excluders (*n*= 436). In France, self-exclusion is a voluntary process; its duration is fixed by the player from 1 day up to a maximum of 3 years. At the end of the self-exclusion period, the gamblers are notified by email by the provider, and they are then allowed to gamble again on the platform without any additional procedure. At no point during the self-exclusion process is guidance or any kind of help offered. Self-exclusion prevents the gambler from any kind of gambling activity on the website during the chosen period of time.

### 2.2. Measures

We collected data retrospectively from different prospective databases systematically recorded by the gambling service provider: (a) Gambler data: self-excluders’ basic demographics (gender, age, date of opening of the account), characteristics of self-exclusions (date, duration) and detailed gambling variables in the month prior to self-exclusion: cash game winnings, prize amounts for tournaments, buy-in-plus-rake for tournaments, session characteristics (starting date, end date, duration). Sessions were defined as gambling with no period under 10 min without action. This measure was based on our clinical experience and on the information provided by the provider of no systematic disconnection when leaving from the website or the application on wireless devices especially. This measure was built on the experience of difficulty in extracting and interpretation of sessions duration when taking into account connection time only in a previous study [14]. We chose to explore 4-week periods because most employed people in France receive their income once a month. It is therefore important to capture at least 4 weeks per period to avoid any artificially enhanced gambling activity resulting from a possible effect following receipt of income.

Money and time spent in the preceding 4 weeks were the 2 outcomes of interest and were defined as follows: (a): time spent was obtained by summing all session durations in the last 4 weeks. Session duration was obtained by subtracting session end date from session starting date. Money spent in the last 4 weeks was defined as the net losses in the previous 4 weeks, obtained from all cash game and tournament gambling data at table level for players using real money. Table net loss was obtained from the reverse of winnings. Winnings were computed from table data (cash game winnings + prize amount for tournaments-buy-in-plus-rake for tournaments). Account-duration was defined as the time between opening the account on the website and the self-exclusion date, or the self-exclusion date of the matched self-excluders for matched gamblers. Money and time spent in the last 4 weeks on poker on the website were calculated at the self-exclusion date (or self-exclusion date of the matched self-excluders for matched gamblers), and at 3, 4, 6 and 12 months after the end of the self-exclusion period (or after self-exclusion date of the matched self-excluders for matched gamblers). We intentionally took the 12 months period after the end of the self-exclusion period into account to explore any possible changes in patterns over time when gambling was again accessible on the website, the self-exclusion period itself being of no interest for the variables studied, since gamblers were prevented from gambling. There was no missing data.

### 2.3. Sub-Groups

As we could not match our sample for gambling involvement, we chose to additionally analyze subgroups with similar levels of involvement in terms of money and time. Sub-groups of the gamblers who were the most heavily involved were defined as follows: gamblers from the highest quartile for amounts of money/time spent in the last 4 weeks, respectively >170 €/23 hours. In this sub-group analysis, the matching ratio of 1:1 could not be maintained, and gamblers could no longer be matched on age, sex and account duration. However, the mean age and the proportion of males remained very close across groups: 32.13 years (*sd* = 9.68) and 86% male among the self-excluders who were the most heavily involved in terms of money (*n* = 2265) vs 33.08 years (*sd* = 10.15) and 86% male among the gamblers who were the most heavily involved in money in the matched group (*n* = 79) and 32.05 years (*sd*= 9.74) and 87% male among the self-excluders who were the most heavily involved in terms of time (*n* = 2150) vs 32.73 years (*sd* = 9.72) and 86% male among the gamblers in the matched group who were the most heavily involved in terms of time (*n* = 185).

Short-duration self-exclusion was defined as <90 days (*n* = 1460). In this group, money and time spent in the last 4 weeks were collected at 4, 6 and 12 months after the start of self-exclusion.

### 2.4. Statistical Analysis

The money and time spent over 12 months after the end of the self-exclusion period were analyzed using a mixed model with the subjects as a random effect. The fixed effects were self-exclusion, time as a categorical variable, and their interaction. This interaction of self-exclusion and time provides a test for the null hypothesis that “the reduction in money/time spent over the 12 months after the end of the self-exclusion would not be different between the two groups”: we report here only the *p*-value of the ANalysis Of Variance (ANOVA) between the mixed models with and without the interaction (i.e., the “null” model), which is in accordance with our hypothesis, testing for an effect of self-exclusion on time/ money spent at any time point over the 12 months. A significant interaction effect means that there are significant differences between groups and over time. In other words, the change in scores over time is different depending on group membership. Analyzes were performed on the whole sample, on the sub-groups with the greatest time or money involvement and on short-duration self-excluders. As sample sizes were smaller in the subgroups of gamblers who were the most heavily involved and led to a lack of power, we completed our analysis with the calculation of effect sizes for self-exclusion at 12 months in these subgroups. We use the Morris *d_2_* which is a standardized measure of effect size suitable for groups with unequal sample sizes within a pre-post-control design [18]. Additionally, we calculated the effect size for short self-exclusions (< 90 days) at month 12 after self-exclusion among the gamblers who were the most heavily involved (respectively in terms of money / time, self-excluders and matched: *n* = 683 and 18/*n =* 665 and 35).The strength of the effect sizes was determined using descriptors of magnitudes of *d* = 0.01 to 2.0, as initially suggested by Cohen and expanded by Sawilowsky [19].

All tests were 2-sided and performed with R software V3.5.1. (R core Team, free collaborative software).

### 2.5. Ethics

Gamblers were informed of, and consented to, personal and gambling data collection and analysis in the general conditions of use when opening an account on the website. Data collection and analysis by Winamax were authorized by the “Comité National Informatique et Libertés” (CNIL) and registered with CNIL declaration n° 1430126, which allows the analysis of the routinely recorded data for public health purposes.

The study respected the STrengthening the Reporting of OBservational studies in Epidemiology (STROBE) statement checklist items [20]. 

## 3. Results

### 3.1. Matched Gamblers

The matched gamblers were aged 31.5 (*sd* = 9.5) on average and predominantly male (87%). Money/time spent by the matched gamblers in the preceding 4 weeks amounted to 12.9€ (*sd* = 145.5)/ 3.5h (*sd* = 12.9) on average before the self-exclusion day of their matched self-excluders, and 4.2€ (*sd* = 72.5)/ 1.3h (*sd* = 7.2) at12 months after the end of the self-exclusion period of their matched self-excluders. Account age was 322.33 days on average (*sd* = 445.83).

### 3.2. Self-Excluders

The characteristics of the first-time self-excluders and short-duration first-time self-excluders are presented in Table 1. The self-excluders were aged 31 on average and predominantly male. The short-duration first-time self-excluders amounted to 30% of all first-time self-excluders over the 7 years. Money/time spent by self-excluders in the last 4 weeks was 398.5€ (*sd* = 1221.4)/ 32.8h (*sd* = 40.1) before self-exclusion, and 32.3€ (*sd*= 386.6)/6.3h (*sd* = 20.0) at12 months after the end of the self-exclusion period. The mean length of self-exclusion was 614 days (*sd* = 499). Short-duration first-time self-excluders were younger and had a greater financial and time involvement in gambling; their account was one month older on average than in the overall sample. 

### 3.3. Effects of Self-Exclusion over 12 Months after the End of the Self-Exclusion Period

A significant effect of self-exclusion was found for money and time spent over the 12 months after the end of the self-exclusion period (*p*-value for both models < 2.2e−16) using mixed models with a subject random effect (Figure 1).

### 3.4. Effect of Self-Exclusion over 12 Months after the End of the Self-Exclusion Period among the Most Heavily Involved Gamblers

The average amount of money spent in the four weeks before and after the self-exclusion period among the gamblers who were the most heavily involved in terms of money, among self-excluders (*n* = 2255) and in the matched group (*n* = 79) are shown in Figure 2. No significant effect of self-exclusions was found on the amounts of money spent (*p* = 0.072) and the effect size was very small (*d* = 0.18). 

The average amount of time spent in the 4 weeks before and after the self-exclusion period among the gamblers who were the most heavily involved in terms of time among self-excluders (*n* = 2150) and in the matched group (*n* = 185) are shown in Figure 3. A significant effect of self-exclusion was found for time spent (*p* < 2.2e−16) and the effect size was small (*d* = 0.34). 

### 3.5. Short Self-Exclusions

Significant effect of short self-exclusion was found for money and time spent over the 12 months after a short self-exclusion (*p*-value in both models <2.2e−16) using mixed models with a subject random effect (Figure 3).

No significant effect of short self-exclusions was found on money/ time spent gambling among the gamblers who were the most heavily involved in terms of money/time (respective *p*-values = 0.873 and 0.491) (Figure 4) but the sizes of the control groups were very small (respectively *n* = 683 vs 18, and *n* = 665 vs 35). The effect size was very small for money spent (*d* = 0.17), and negative and below the very small level for time spent (*d* = −0.09). 

## 4. Discussion

This is the first real life study, reporting comparative follow-up data on voluntary self-exclusion on the initiative of gamblers and including non-self-selected gamblers. This retrospective study analyzed prospectively registered account -based gambling data. The aim was to assess the efficacy of self-exclusion in the long term in term of time and money involvement. 

The analysis of account-based gambling data for all first-time self-excluders on a website over 7 years confirmed the efficacy of self-exclusion on gambling outcomes in the long term. The exhaustiveness of this data is a strength that ensures representativeness and power for the statistical analyzes. However, the effect of self-exclusion among the most heavily involved gamblers was found only for the time spent, and not for the money spent, despite a very high level of expenditure before self-exclusion in this subgroup [14]. One important piece of information here is the spontaneous decrease in gambling involvement among gamblers who were the most heavily involved and who did not self-exclude. This result is congruent with a high rate of spontaneous remissions observed in gambling disorder [21]. This result shows the need to provide comparative data, more informative than a tool that is de facto considered to be efficient and promoted by the regulatory authorities [17]. Another interpretation of this decrease among heavy gamblers who did not self-exclude is that the gamblers were not randomized here, and could have chosen to self-exclude if they lacked confidence in their ability to bring about a change in their gambling without an external constraint such as self-exclusion, the reverse being true for non-self-excluders. The efficacy of short self-exclusions among the most heavily involved gamblers was not supported by our data. This is in line with recent experimental data among problem gamblers suggesting no efficacy of very short self-exclusions on gambling outcomes [10]. Another qualitative study reported a preferred duration to ensure efficacy of 12 months from the perspective of problem gamblers who self-excluded [22].

This study presents some important limitations. First, we included only poker gamblers. As discussed in the introduction, poker gamblers present particular cognitive profiles. Moreover, online poker gamblers are younger than other gamblers [4,12], and their history of gambling and associated damages could differ, as well as their motivation to change. The presented results could reflect some of these particularities and not be true in other gambling activities. No data was available on gambling on other online or offline gambling service providers. Gamblers could have just switched from one website to another during the exclusion period. However, all follow-up data reported here concerns gambling after the end of the exclusion period. Gamblers can gamble back on the website after this period and are commercially encouraged and sometimes offered incentives to do so. Moreover, we have already documented in another study that most gamblers return to the initial website to gamble after a self-exclusion [1]. On the other hand, as gambling is regulated in France, gamblers have to provide their Identity Card when opening an account; this measure theoretically prevents from gambling from an account opened under a false identity. The gambling profiles observed are still informative as such, even if not representative of all gambling activities. The use of account-based gambling data is a strength of the study because it enables objective data to be reported. However, it would be interesting to document the effect of self-exclusion on non-gambling outcomes, such as quality of life. No formal diagnosis of gambling disorder and no information on mental disorders or comorbidities were available. We could not, for technical reasons, match the gamblers for the level of gambling involvement in terms of time and money. Our statistical analysis allows for comparisons between the groups by adjusting the mixed model on the subject, which takes into account all subject characteristics including gambling involvement; however, it does not replace a control group with similar involvement in gambling in term of time and money spent. In addition, self-excluders could be different from non-self-excluders in term of the degree of motivation to change, as self-exclusion is a voluntary process in France. Finally, the naturalistic and ecological design of this study of course prevented any randomization process. We therefore report here results on the effect of self-exclusion rather than on efficacy. Further studies could inform on possible response factors to self-exclusion. No information was available on the health care resources used by the gamblers included.

## 5. Conclusions

Self-exclusion seems efficient in the long term (i.e., 12 months after the end of the self-exclusion period). However, the effects on money spent as a result of self-exclusion or short self-exclusion should be further explored among the online poker gamblers who are the most heavily involved. A spontaneous, clinically-relevant decrease in gambling activities was demonstrated among most involved gamblers who did not self-exclude. Further study with a randomized design and non-gambling outcomes should be conducted to conclude robustly on the efficacy of short and long self-exclusions in problem gambling, and on response factors.

## Figures and Tables

**Figure 1 ijerph-16-04399-f001:**
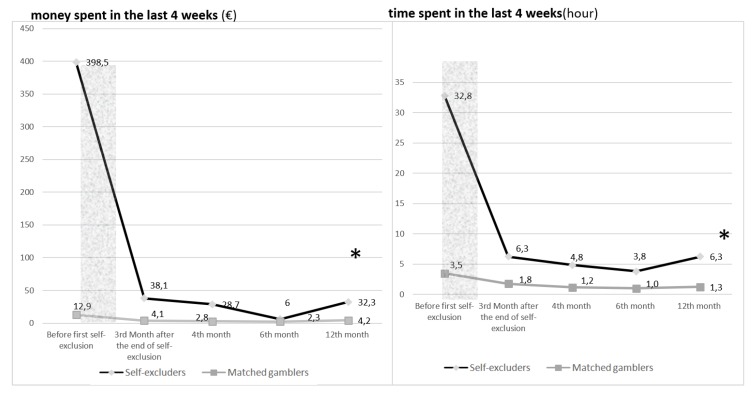
Evolution of money/time spent in the last 4 weeks (€/hours) at baseline and after the end of self-exclusion period (*n* = 4887 and *n* = 4451). (* = *p*-value < 0.05—ANOVA between the mixed model with and the null model without the interaction of self-exclusion X time).

**Figure 2 ijerph-16-04399-f002:**
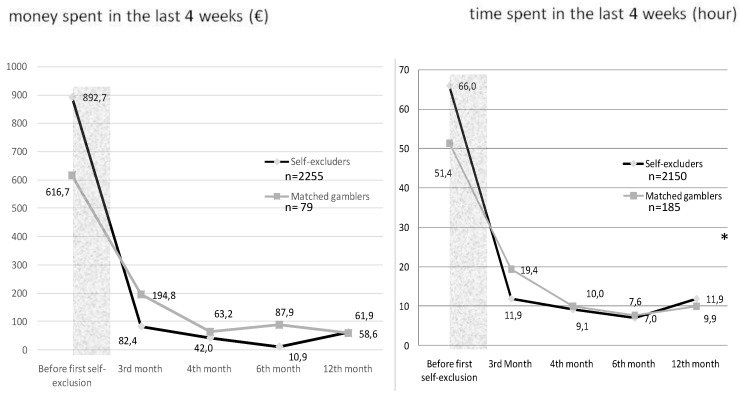
Evolution of money spent (net loss) in the last 4 weeks before and after the self-exclusion period among the gamblers who were the most heavily involved in terms of money (*n* = 2255 and 79 respectively for the self-excluders and the control group of matched gamblers) and time (*n* = 2150 and 185 respectively for the self-excluders and the control group of matched gamblers) (* = *p-*value < 0.05 —ANOVA between the mixed model with and the null model without the interaction of self-exclusion X time). (* = *p-value < 0.05* - ANOVA between the mixed model with and the null model without the interaction of self-exclusion X time).

**Figure 3 ijerph-16-04399-f003:**
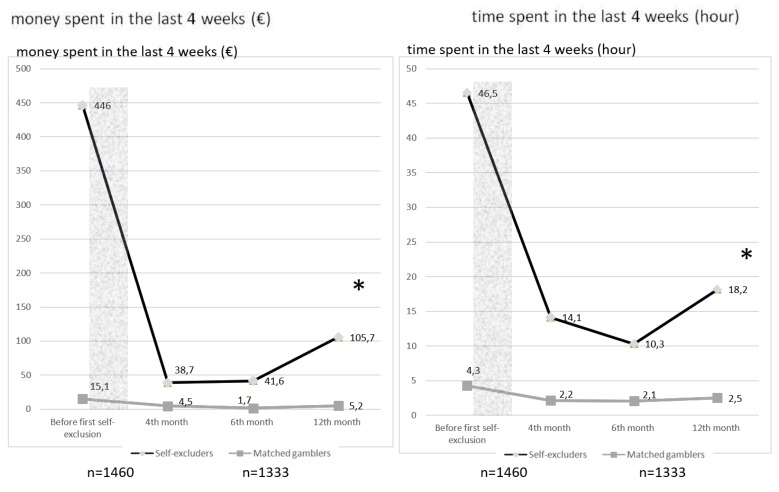
Evolution of money / time spent in the last 4 weeks (€/hours) before and after a short self-exclusion (*n* = 1460 and 1333). (* = *p*-value < 0.05—ANOVA between the mixed model with and the null model without the interaction of self-exclusion X time).

**Figure 4 ijerph-16-04399-f004:**
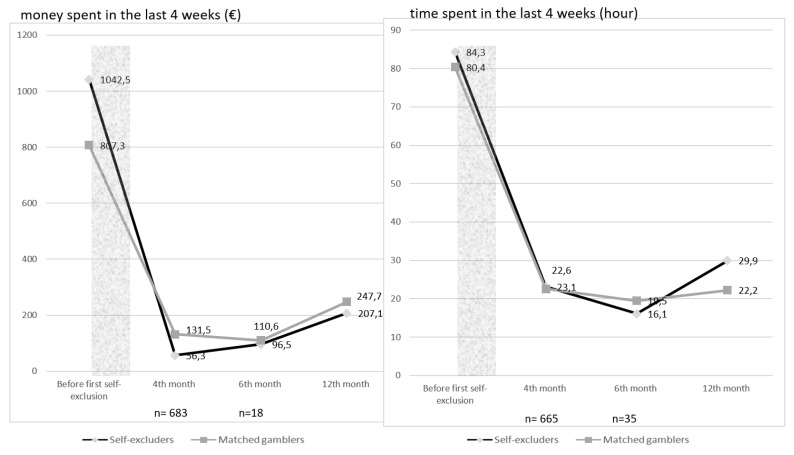
Evolution of money spent (net loss) in the last 4 weeks before and after a short self-exclusion among the gamblers who were the most heavily involved in terms of money (*n* = 683 and 18 respectively for the self-excluders and the control group of matched gamblers) and in terms of time (*n* = 665 and 35 respectively for the self-excluders and the control group of matched gamblers). (* = *p*-value < 0.05—ANOVA between the mixed model with and the null model without the interaction of self-exclusion X time).

**Table 1 ijerph-16-04399-t001:** Characteristics of first-time self-excluders, and short-duration first-time self-excluders subgroup.

Characteristics	All 1^st^ Self-Exclusions *n* = 4887	Short Self-Exclusions *n* = 1460
Account age (days), mean (sd)	272.4 (407.1)	307.59 (415.63)
Age (years), mean (sd)	31.4 (9.6)	30.48 (9.21)
Gender(male), n(%)	4252 (87.0)	1270 (87.99)
Money spent in the last 4 weeks before self-exclusion (€), mean (sd)	398.5 (1221.4)	445.96 (1350.21)
Time spent in the last 4 weeks before self-exclusion (minutes), mean (sd)	1969.8 (2406.0)	2791.3 (2706.4)
Self-exclusion period duration, days, mean (sd)	614.0(499.0)	32.0 (23.0)

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
