# Peer review of "Self-Exclusion among Online Poker Gamblers: Effects on Expenditure in Time and Money as Compared to Matched Controls"

_ijerph, 2019, doi:10.3390/ijerph16224399_

Round 1
Reviewer 1 Report
This is an interesting paper, but I have several concerns with the current draft. As outlined below.
Lines 55 Authors cited a study using follow-up data in line 58-60. In line 227 authors mentioned the first study. Please clarify the difference from the previous study.
Line 87 What is the evidence for this measure? It would be great to know further details.
Line 112-114 These sentences would be confusing.
Line 120-121 Please provide more evidence.
Statistical Analysis
In the case of short self-exclusions, there is a large difference in the group sample size. My concern is whether it is appropriate to verify statistical significance with ANOVA.
I suggest to carefully refer to the most recent literature in the self-exclusion filed. Please consider the following suggestions:
Caillon, J., Grall-Bronnec, M., Perrot, B., Leboucher, J., Donnio, Y., Romo, L., & Challet-Bouju, G. (2019). Effectiveness of At-Risk Gamblers’ Temporary Self-Exclusion from Internet Gambling Sites. Journal of gambling studies, 35(2), 601-615.
Kotter, R., Kräplin, A., Pittig, A., & Bühringer, G. (2019). A systematic review of land-based self-exclusion programs: Demographics, gambling behavior, gambling problems, mental symptoms, and mental health. Journal of gambling studies, 35(2), 367-394.
Bonello, M., & Griffiths, M. D. (2019). Behavioural tracking, responsible gambling tools, and online voluntary self-exclusion: implications for the gambling industry. Casino and Gaming International, 38, 41-45.
Author Response
Response to reviewer 1:
We are very grateful to the reviewers for their comments, which gave us the opportunity to improve our manuscript. Please find above the detailed responses to their comments
Reviewer 1:
This is an interesting paper, but I have several concerns with the current draft. As outlined below.
Lines 55 Authors cited a study using follow-up data in line 58-60. In line 227 authors mentioned the first study. Please clarify the difference from the previous study.R1: We apologize for this inconsistency. This is the first comparative real life study with non self-selected gamblers who self-excluded. We corrected the sentence line 224: “first real life study reporting comparative follow-up data on voluntary self-exclusion on the initiative of gamblers and including non self-selected gamblers”
Line 87 What is the evidence for this measure? It would be great to know further details.
R2. This is a measure based on our clinical experience and on the information provided by the provider of no systematic disconnection when leaving from the website or the application on wireless devices especially. This measure was built on the experience of difficulty in extracting and interpretation of sessions duration when taking into account connection time only in a previous study. We added this details in the manuscript: “This measure was based on our clinical experience and on the information provided by the provider of no systematic disconnection when leaving from the website or the application on wireless devices especially. This measure was built on the experience of difficulty in extracting and interpretation of sessions duration when taking into account connection time only in a previous study.”
Line 112-114 These sentences would be confusing.R3. We apologize for the complexity of this paragraph, but we thought it was important to provide the reader the detail of financial data construction, in order to illustrate the transparency of our data extraction and analysis. We would like to keep this paragraph this way.
Line 120-121 Please provide more evidence.This approach is well explained here: https://www.statisticssolutions.com/statistical-interaction-more-than-the-sum-of-its-parts/ We added the following sentence to clarify this common approach. “A significant interaction effect means that there are significant differences between groups and over time. In other words, the change in scores over time is different depending on group membership.”
Statistical Analysis
In the case of short self-exclusions, there is a large difference in the group sample size. My concern is whether it is appropriate to verify statistical significance with ANOVA.R4. We respectfully would like to remind the reviewer that our model is a mixed effects model, appropriate for unequal sample sizes then, and that the anova is used here to obtain a p-value for the interaction term from the mixed model. This is not an ANOVA between the two groups.
I suggest to carefully refer to the most recent literature in the self-exclusion filed. Please consider the following suggestions:Caillon, J., Grall-Bronnec, M., Perrot, B., Leboucher, J., Donnio, Y., Romo, L., & Challet-Bouju, G. (2019). Effectiveness of At-Risk Gamblers’ Temporary Self-Exclusion from Internet Gambling Sites. Journal of gambling studies, 35(2), 601-615.
Kotter, R., Kräplin, A., Pittig, A., & Bühringer, G. (2019). A systematic review of land-based self-exclusion programs: Demographics, gambling behavior, gambling problems, mental symptoms, and mental health. Journal of gambling studies, 35(2), 367-394.
Bonello, M., & Griffiths, M. D. (2019). Behavioural tracking, responsible gambling tools, and online voluntary self-exclusion: implications for the gambling industry. Casino and Gaming International, 38, 41-45.
We respectfully would like to mention that the first reference is already included in our manuscript. The second one regards self-exclusion from land based environment, whereas our study is about online self-exclusion from online environment. The third one was not published at the time of submission of our manuscript, so we could not have included it. As this is an expert opinion with no new data, we don’t judge it necessary to include this reference in our manuscript.
Reviewer 2 Report
This is a well written paper that presents a careful temporal analysis of player data from sites offering online poker. The matched sampling methodology seems sound, the N is large - due to access being provided by operators - the limitations are spelt out, and the conclusions are certainly justified from the results. The contribution is sound and useful, particularly in providing a matched group so as to compare changes between self-excluders and spontaneous remission among the matched group.
I have no major criticisms, and therefore no hesitation in recommending for publication. As it's carefully written and presented, I have few minor points to note. However, the missing spaces throughout should be checked by the typesetter.
line 69 - and1:1 - space req
line 74 - 'doubloons' - is this a technical term for double matches I haven't heard of?
line 76 - exclusionperiod - space
line 77, 78 - theprovider, nopoint - space I will stop mentioning every time this occurs - please check whole document
line 136 - dppc2sensu - what is 'sensu'?
Author Response
Responses to reviewer 2
We are very grateful to the reviewers for their comments, which gave us the opportunity to improve our manuscript. Please find above the detailed responses to their comments
Reviewer 2
This is a well written paper that presents a careful temporal analysis of player data from sites offering online poker. The matched sampling methodology seems sound, the N is large - due to access being provided by operators - the limitations are spelt out, and the conclusions are certainly justified from the results. The contribution is sound and useful, particularly in providing a matched group so as to compare changes between self-excluders and spontaneous remission among the matched group.
I have no major criticisms, and therefore no hesitation in recommending for publication. As it's carefully written and presented, I have few minor points to note. However, the missing spaces throughout should be checked by the typesetter.line 69 - and1:1 - space req
line 74 - 'doubloons' - is this a technical term for double matches I haven't heard of?
line 76 - exclusionperiod - space
line 77, 78 - theprovider, nopoint - space I will stop mentioning every time this occurs - please check whole document
We thank the reviewer and apologize for the missing spaces which were due to a conflict in the used office word version. Corrections have been made.
line 136 - dppc2sensu - what is 'sensu'?
R2. The mention sensu can be removed. We removed it from the manuscript.